# FROM GENERAL TO EXPERT: CUSTOM PRUNING LLMS ACROSS LANGUAGE, DOMAIN, AND TASK

## ABSTRACT

Large Language Models (LLMs) have transformed natural language processing, yet their substantial model sizes often demand significant computational resources. To conserve computing resources and increase inference speed, it is crucial to prune redundant parameters, especially for general users who often need expert models tailored to specific downstream scenarios. However, current pruning methods primarily focus on maintaining models' general capabilities, either requiring extensive post-training or performing poorly due to coarse-grained pruning. In this work, we design a Custom Pruning method (`Cus-Prun`) to prune a large general model into a smaller expert model for specific scenarios. `Cus-Prun` positions an expert model along the "language", "domain" and "task" dimensions. By identifying and pruning irrelevant neurons, it creates expert models without any post-training. Our experiments demonstrate that `Cus-Prun` consistently outperforms other methods, achieving minimal loss in both expert and general capabilities across various models from different model families and sizes.

## 1 INTRODUCTION

Large Language Models (LLMs) (Achiam et al., 2023; Reid et al., 2024; Dubey et al., 2024; Team et al., 2024) have revolutionized the field of natural language processing, emerging as powerful tools with widespread applications across various languages (Cui et al., 2023; Yang et al., 2024a), domains (Li et al., 2023a; Roziere et al., 2023; Li et al., 2023b), and tasks (Azerbayev et al., 2024; Alves et al., 2024). However, the impressive performance of LLMs often comes at the cost of immense model sizes, mostly containing billions of parameters and thus demand significant computing resources (Goldstein et al., 2023; Musser, 2023). To address this issue, researchers have recently proposed various pruning methods for LLMs. These methods aim to reduce model parameters while maintaining the model's overall performance through techniques such as removal of unimportant structures (Men et al., 2024; Song et al., 2024; Zhang et al., 2024; Ma et al., 2023), matrix approximation (Zhao et al., 2024a; Sharma et al., 2024; Ashkboos et al., 2024), and extensive post-training after pruning (Wang et al., 2024; Xia et al., 2024).

These existing pruning methods have primarily focused on preserving the *general capabilities* of the model, often evaluated using compound benchmarks such as MMLU (Hendrycks et al., 2021) consisting of a broad spectrum of tasks. While aiming for overall versatility, they may not align well with real-world user needs, which are usually more *specific and targeted*. For instance, a user might require a question-answering model tailored specifically for the education domain in German. Such specialized request aligns closely with the fundamental motivation behind pruning: to create a smaller model by eliminating unnecessary parameters. In this context, "unnecessary" becomes much clearer—parameters that are irrelevant to the specific use case can be considered redundant. Pruning could therefore be leveraged to remove parameters irrelevant to the target language, domain, or task, thereby producing a more specialized expert model for the desired application. However, current pruning techniques primarily focus on general capabilities, especially for traditional NLP tasks in English, and often employ coarse-grained pruning approaches, and sometimes require extensive post-training after pruning (Xia et al., 2024; Zhao et al., 2024a; Men et al., 2024; Zhang et al., 2024). Therefore, a more fine-grained and expert model targeting approach is needed to effectively tailor models to particular user needs while maintaining the general performance.

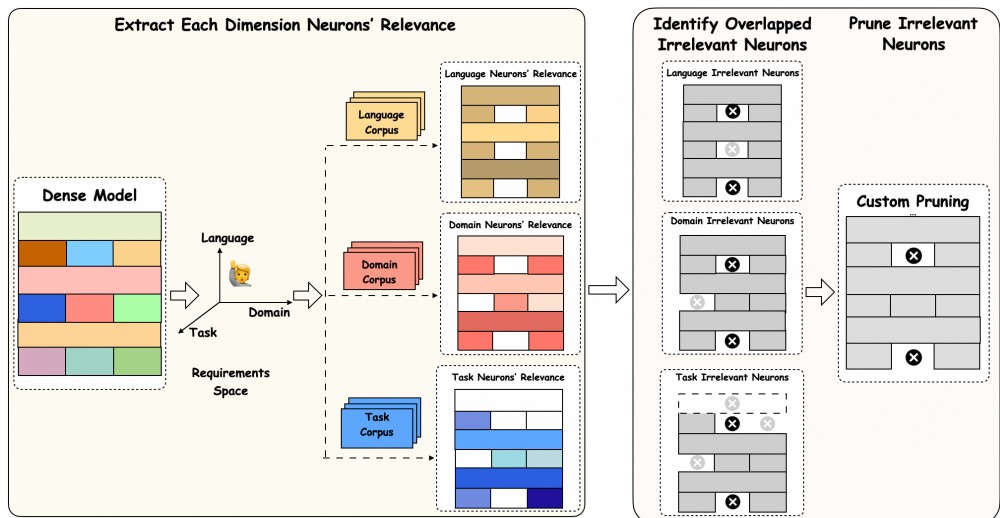

Figure 1: Given a request for an expert model across three dimensions (language, domain, task), `Cus-Prun` (i) identifies irrelevant neurons for each dimension with the corresponding corpus, and (ii) prunes overlapped irrelevant neurons across dimensions to obtain the expert model.

In this work, we introduce a novel Custom Pruning (`Cus-Prun`) method, designed to prune a large general model into a small specialized expert model tailored for specific scenarios. We first define the expert model by positioning the target user's needs along three key dimensions: language (e.g., English, Chinese, Germain), domain (e.g., E-commerce, education), and task (e.g., QA, summarization). Then motivated by existing studies that certain neurons are responsible for certain functions (Zhao et al., 2024b; Tang et al., 2024; Liang et al., 2024), `Cus-Prun` identifies and preserves critical neurons that are more relevant to particular languages, domains, or tasks, while pruning less relevant ones, ultimately leading to a smaller expert models. Specifically, as illustrated in Figure 1, `Cus-Prun` involves two main steps: First, it identifies irrelevant neurons for each dimension by assessing the impact of their removal on the generated output when processing corresponding corpus. A neuron is deemed irrelevant if zeroing its parameters affects the output by a specified margin. Such corpus for each dimension could be easily constructed from the relevant plain text documents. Next, we construct the expert model by pruning common irrelevant neurons across all dimensions. Therefore, it allows for the creation of expert models that excel in specific scenarios, such as the German QA model in the education domain, without the need for extensive post-training or fine-tuning. Furthermore, `Cus-Prun`'s flexibility allows it to focus on one, two, or all three dimensions (language, domain, task) as needed, making it adaptable to a wide range of real-world applications where specialized LLMs are required.

We conduct comprehensive experiments to evaluate the performance of `Cus-Prun` across various scenarios. Experimental results demonstrate that it consistently outperforms other pruning methods in all settings. For three-dimensional specific expert models, `Cus-Prun` prunes 25.0% of parameters while incurring only a 14% drop in expert capability (averaging across multilingual, multidomain, and multitask datasets) and 12% on general capability (averaging performance on three representative compound NLP benchmarks) for Llama2-13B. In contrast, others suffer a 38% reduction in expert capabilities and a 29% decline in general capabilities. This trend is consistent across multiple models from different model families and sizes, such as Mistral-Nemo, Llama3-8B, and Llama3-70B. For more focused applications, such as two- or one-dimensional specific expert models (e.g., language-domain specific or language-specific models), `Cus-Prun` also surpasses other pruning methods, demonstrating its versatility and effectiveness across various specialized settings.

## 2 CUSTOM PRUNING (CUS-PRUN)

An expert model could be generally positioned from three dimensions: "language" ($L \in \mathbb{L}$), "domain" ($D \in \mathbb{D}$), and "task" ($T \in \mathbb{T}$), which can be represented as $LLM_{\text{Exp}} := (L, D, T) \in \mathbb{L} \times \mathbb{D} \times \mathbb{T}$. Specifically, the language dimension encompasses various languages such as English, Spanish, and

Thai. The domain dimension covers different fields like finance, legal, and medical. The task dimension includes various applications such as question-answering, data-to-text, and summarization.

As expert models focus solely on specific capabilities, some unused capabilities such as support for irrelevant languages or domains will inevitably become redundant. To optimize computing resources and increase inference speed, we could prune redundant parameters that do not align with our current objectives. In this section, we propose a custom pruning method named `Cus-Prun` to derive expert models with flexible customization granularity.

## 2.1 FOUNDATIONAL CUSTOM PRUNING

Drawing inspiration from recent neuron interpretation studies (Tang et al., 2024; Liang et al., 2024; Zhao et al., 2024b) that many parameters in the model are irrelevant to processing a specific "language", we hypothesize that this phenomenon can be extended to other dimensions such as "domain" and "task", meaning that certain parameters remain unused when handling a specific dimension. In contrast to other studies that examine redundant layers (Song et al., 2024; Men et al., 2024) or modules (Zhang et al., 2024), `Cus-Prun` involves a more fine-grained investigation focusing on redundant neurons, which are defined as individual rows or columns within the parameter matrix of a language model. Concretely, when handling each dimension, we identify a specific set of *irrelevant neurons* in the original LLM, denoted as $\tilde{\mathcal{N}}_L$, $\tilde{\mathcal{N}}_D$, and $\tilde{\mathcal{N}}_T$ for $L$, $D$, and $T$, respectively. An expert LLM can be obtained by removing neurons that are irrelevant to all three dimensions. Specifically, to identify irrelevant neurons corresponding to the selected dimension, we construct a corpus within that dimension while ablating others. For example, to determine irrelevant neurons for a specific language $L_{\text{Exp}}$, we create a corpus set $C_{L_{\text{Exp}}} = \{(L_{\text{Exp}}, D, T) | D \in \mathbb{D}, T \in \mathbb{T}\}$, comprising documents in language $L_{\text{Exp}}$ across various domains $D$ and tasks $T$. We then identify neurons that are irrelevant across all documents in $C_{L_{\text{Exp}}}$, i.e.,

$$\tilde{\mathcal{N}}_{L_{\text{Exp}}} = \big\{ \text{Neuron} \big| \text{Irrelevant to } c, \text{for all } c \in C_{L_{\text{Exp}}} \big\}, \tag{1}$$

where a neuron is considered irrelevant if its removal, by setting its parameters to zero, affects the generated output below a specified threshold.

Specifically, we denote the $l$-th neuron in layer $i$ as $N_i^{(l)}$, and the intermediate representation after layer $i$ when handling document $c \in C_{L_{\text{Exp}}}$ as $h_i(c)$. The degree of relevance of neuron $N_i^{(l)}$ in processing $c$ is calculated by $\|h_{\backslash N_i^{(l)}, i}(c) - h_i(c)\|_2$, where $h_{\backslash N_i^{(l)}, i}(c)$ represents the intermediate representation after deactivating neuron $N_i^{(l)}$. Therefore, the irrelevant neurons of the model when handling document $c$ is the set

$$\tilde{\mathcal{N}}_c = \{ N_i^{(l)} \big| \|h_{\backslash N_i^{(l)}, i}(c) - h_i(c)\|_2 \leq \epsilon, \text{ for all } N_i^{(l)} \text{ in } \mathcal{LLM} \}, \tag{2}$$

where $\epsilon$ is a pre-defined threshold.

Therefore, Equation 1 is equivalent to

$$\tilde{\mathcal{N}}_{L_{\text{Exp}}} = \{ N_i^{(l)} \big| N_i^{(l)} \in \tilde{\mathcal{N}}_c, \text{ for all } c \in C_{L_{\text{Exp}}} \text{ and } N_i^{(l)} \text{ in } \mathcal{LLM} \}. \tag{3}$$

Similarly, we establish corresponding corpus sets for other dimensions, $C_{D_{\text{Exp}}} = \{(L, D_{\text{Exp}}, T) | L \in \mathbb{L}, T \in \mathbb{T}\}$ and $C_{T_{\text{Exp}}} = \{(L, D, T_{\text{Exp}}) | L \in \mathbb{L}, D \in \mathbb{D}\}$, to extract irrelevant neurons, $\tilde{\mathcal{N}}_{D_{\text{Exp}}}$ and $\tilde{\mathcal{N}}_{T_{\text{Exp}}}$. Finally, the expert model is constructed by

$$\mathcal{LLM}_{\text{Exp}} := \mathcal{LLM} \setminus \{\tilde{\mathcal{N}}_{L_{\text{Exp}}} \cap \tilde{\mathcal{N}}_{D_{\text{Exp}}} \cap \tilde{\mathcal{N}}_{T_{\text{Exp}}}\}. \tag{4}$$

## 2.2 ADAPTIVE CUSTOM PRUNING

Besides three-dimensional expert models, requirements involving constraints in one or two dimensions are also common in real-world applications (Roziere et al., 2023; Alves et al., 2024). For instance, a language-specific model or a domain-specific model is one-dimensional, whereas a language-domain-specific model (such as a Chinese Medical LLM) constrains two dimensions. Therefore, in this section, we extend `Cus-Prun` to prune expert models in different granularities.

---

**Algorithm 1** Adaptive Custom Pruning

---

**Input:** Request for expert model $LLM_{Exp}$ with selected dimensions: $L_{Exp}, D_{Exp}, T_{Exp}$ (any subset).
1: // Construct specific corpora for each selected dimension.
2: $C = \{\}$
3: **if** $L_{Exp}$ is specified **then**
4:     $C = C \cup \{(L_{Exp}, D, T) \mid D \in \mathbb{D}, T \in \mathbb{T}\}$
5: **end if**
6: **if** $D_{Exp}$ is specified **then**
7:     $C = C \cup \{(L, D_{Exp}, T) \mid L \in \mathbb{L}, T \in \mathbb{T}\}$
8: **end if**
9: **if** $T_{Exp}$ is specified **then**
10:     $C = C \cup \{(L, D, T_{Exp}) \mid L \in \mathbb{L}, D \in \mathbb{D}\}$
11: **end if**
12: // Identify irrelevant neurons for each selected dimension.
13: **for all** $neuron$ in $\mathcal{LLM}$ **do**
14:     **if** $\forall c \in C, neuron$ is not relevant to $c$ **then**
15:         Add $neuron$ to the set of irrelevant neurons $\tilde{\mathcal{N}}$
16:     **end if**
17: **end for**
18: // Prune irrelevant neurons to obtain expert model.
19: $\mathcal{LLM}_{Exp} = \mathcal{LLM} \setminus \tilde{\mathcal{N}}$
**Output:** $\mathcal{LLM}_{Exp}$

---

**Two-Dimensional Specific Expert Model**   Without losing generality, we use the language-domain expert model as a concrete example, which requires an expert model constrained in two dimensions: language ($L_{Exp}$) and domain ($D_{Exp}$). We derive the sets of irrelevant neurons $\tilde{\mathcal{N}}_{L_{Exp}}$ and $\tilde{\mathcal{N}}_{D_{Exp}}$ according to Equation 3. We obtain the expert model by pruning the original dense model as follows:

$$\mathcal{LLM}_{Exp} := \mathcal{LLM} \setminus \{\tilde{\mathcal{N}}_{L_{Exp}} \cap \tilde{\mathcal{N}}_{D_{Exp}}\}. \tag{5}$$

**One-Dimensional Specific Expert Model**   We use the language-specific expert model as an example, which focuses exclusively on optimizing performance for a certain language ($L_{Exp}$), irrespective of domain or task. Similarly, we obtain the language-specific corpus $C_{L_{Exp}}$, then identify irrelevant neurons $\tilde{\mathcal{N}}_{L_{Exp}}$ according to Equation 3, and extract the expert model by

$$\mathcal{LLM}_{Exp} := \mathcal{LLM} \setminus \{\tilde{\mathcal{N}}_{L_{Exp}}\}. \tag{6}$$

The overall algorithm is further detailed in Algorithm 1. To enhance efficiency, we implement the parallel neuron-detection method (Zhao et al., 2024b), which accelerates the sequential calculations from line14 to line16 in Algorithm 1.

## 3   PRELIMINARY EVALUATION

In this section, we conduct preliminary experiments to obtain an expert model that is specific in all three dimensions. This approach can be considered as the most fine-grained operation for developing coarse-grained expert models that are specific in one or two dimensions.

**Experiment Design**   To verify the effectiveness of Cus-Prun in obtaining expert models for specific use cases, we select three datasets corresponding to different user needs: *Korean-Legal-Summarization* (Hwang et al., 2022), *English-Medical-Multiple Choice Questions* (García-Ferrero et al., 2024), and *Chinese-E-commerce-Sentiment Analysis* (Zhang et al., 2015), each named according to the pattern language-domain-task. Then for each scenario, we need to curate the corresponding corpus for each dimension. This curation can be done through manual collection or by automatically retrieving relevant documents online. In our preliminary study, without loss of generality, we employ a strong proprietary model[1] to generate a corpus containing 50 documents for each dimension. The

---

[1] https://platform.openai.com/docs/models/gpt-4o

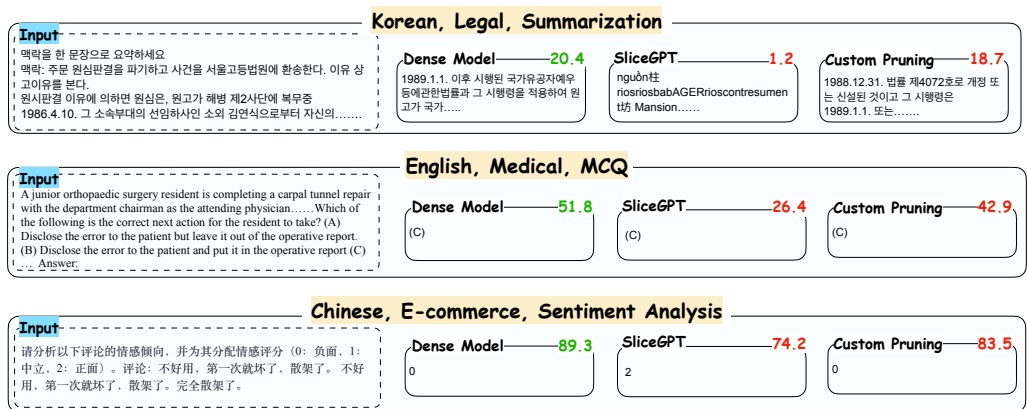

Figure 2: Concrete examples of applying `Cus-Prun` to prune 25% of Llama3-8B-Base's parameters into three-dimensional expert models. Numbers above each box indicate performance on the **whole** test set, with Korean-Legal-Summary evaluated by Rouge-L, and the other two by accuracy.

generated documents could then be used to detect and determine the relevance of neurons for each dimension of each scenario.

**Experiment Setup** We utilize Llama3-8B-Base (Dubey et al., 2024) as the original dense model and set the pruning ratio at 25%. Performance is evaluated using Rouge-L (Lin, 2004) for Korean-Legal-Summary and accuracy score for another two tasks. For comparison, we employ SliceGPT (Ashkboos et al., 2024) as the baseline which replaces each weight matrix with a smaller dense matrix.

**Main Results** Figure 2 presents the results and one concrete example for the original dense model, pruned model with SlideGPT, and pruned model with our proposed `Cus-Prun` method for three distinct use cases. We can observe that `Cus-Prun` largely preserves the performance of the dense model, retraining 92%, 83%, and 94% of the original dense model performance on these three cases respectively. In contrast, the baseline method SliceGPT which does not consider specific use cases largely underperforms compared to `Cus-Prun`. Overall, the results demonstrate that our proposed `Cus-Prun` method could effectively obtain expert models tailored to specific use cases across different languages, domains, and tasks that maintain high performance despite substantial pruning.

## 4 FOUNDATIONAL CUSTOM PRUNING ASSESSMENT

As demonstrated by preliminary evaluation in Section 3, `Cus-Prun` enables the creation of expert language models tailored to specific languages, domains, and tasks. However, when attempting a more comprehensive evaluation, we find that benchmark datasets may not always be available and it is difficult to conduct systematic evaluation. To simplify our evaluation without losing generality, we use two distinct corpora: one focusing independently on a single dimension and another encompassing the remaining two dimensions. This approach allows us to evaluate `Cus-Prun`'s performance in *multilingual*, *multidomain*, and *multitask* settings.

Formally, in the multilingual setting, instead of constructing $C_{L_{\text{Exp}}}$, $C_{D_{\text{Exp}}}$ and $C_{T_{\text{Exp}}}$ independently, we can construct two corpora, $C_{L_{\text{Exp}}}$ and $C_{(D,T)_{\text{Exp}}}$, where $C_{L_{\text{Exp}}}$ helps to identify irrelevant neurons in a specific language ($\tilde{\mathcal{N}}_{L_{\text{Exp}}}$) and $C_{(D,T)_{\text{Exp}}}$ helps to identify irrelevant neurons in a specific domain-task combination ($\tilde{\mathcal{N}}_{D_{\text{Exp}} \cap T_{\text{Exp}}}$). Formally speaking, `Cus-Prun` in Equation 4 is transferred to

$$\mathcal{LLM}_{\text{Exp}} = \mathcal{LLM} \setminus \{\tilde{\mathcal{N}}_{L_{\text{Exp}}} \cap (\tilde{\mathcal{N}}_{D_{\text{Exp}}} \cap \tilde{\mathcal{N}}_{T_{\text{Exp}}})\} \equiv \mathcal{LLM} \setminus \{\tilde{\mathcal{N}}_{L_{\text{Exp}}} \cap \tilde{\mathcal{N}}_{D_{\text{Exp}} \cap T_{\text{Exp}}}\}. \tag{7}$$

Note that this simplification is also applicable to $C_{D_{\text{Exp}}}$, $C_{(L,T)_{\text{Exp}}}$ and $C_{T_{\text{Exp}}}$, $C_{(L,D)_{\text{Exp}}}$.

Table 1: Main Results of `Cus-Prun` on multilingual setting with a pruning ratio of 25%, where "general capability" is tested in English and averaged across several expert models, while "specific capability" is averaged across languages. Results are expressed in Rouge-L in XLSum and in accuracy (%) for other datasets. All models are base models.

| Model | Method | General Capability | | | | Expert Capability | | | | |
|---|---|---|---|---|---|---|---|---|---|---|
| | | ARC-c | GSM8K | MMLU | Avg. | MGSM | M3Exam | XQuAD | XLSum | Avg. |
| Llama3-8B | Dense | 70.7 | 58.3 | 63.1 | 64.1 | 41.2 | 49.1 | 63.4 | 32.9 | 46.7 |
| | LLMPrun. | 26.3 | 2.5 | 24.2 | 17.7 | 1.1 | 24.0 | 13.6 | 23.2 | 15.5 |
| | SliceGPT | 41.5 | 0.0 | 24.2 | 21.9 | 0.0 | 14.9 | 16.6 | 8.5 | 10.0 |
| | ShortGPT | 38.3 | 0.0 | 28.6 | 22.3 | 0.0 | 26.9 | 0.0 | 2.7 | 7.4 |
| | Cus-Prun | **60.3** | **31.9** | **52.1** | **48.1** | **30.1** | **41.5** | **52.6** | **31.5** | **38.9** |
| Mistral-12B | Dense | 82.6 | 68.5 | 50.4 | 67.2 | 51.7 | 43.8 | 49.2 | 25.4 | 42.5 |
| | LLMPrun. | 22.5 | 2.7 | 30.7 | 18.6 | 2.1 | 27.8 | 19.0 | **23.2** | 18.0 |
| | SliceGPT | 49.4 | 1.9 | 32.1 | 27.8 | 0.8 | 25.1 | 17.4 | 7.8 | 12.8 |
| | ShortGPT | 37.8 | 0.0 | 33.9 | 23.9 | 2.9 | 27.0 | 18.0 | 5.0 | 13.2 |
| | Cus-Prun | **67.0** | **39.6** | **43.4** | **50.0** | **34.3** | **39.2** | **40.7** | 23.1 | **34.3** |
| Llama2-13B | Dense | 50.3 | 31.4 | 53.4 | 45.1 | 17.5 | 30.4 | 44.1 | 24.9 | 29.2 |
| | LLMPrun. | 22.4 | 2.1 | 23.6 | 16.0 | 1.1 | 22.8 | 3.8 | 17.7 | 11.3 |
| | SliceGPT | 45.9 | 2.4 | 48.7 | 32.3 | 2.8 | 25.3 | 23.4 | 9.9 | 15.5 |
| | ShortGPT | 39.5 | 3.8 | 37.2 | 26.8 | 2.4 | 23.0 | 24.7 | 11.3 | 15.3 |
| | Cus-Prun | **48.0** | **20.5** | **50.8** | **39.8** | **12.7** | **26.2** | **34.2** | **24.1** | **24.3** |
| Llama3-70B | Dense | 84.1 | 82.7 | 78.8 | 81.9 | 69.5 | 71.1 | 69.1 | 36.6 | 61.6 |
| | LLMPrun. | 69.1 | 26.0 | 53.2 | 49.4 | 16.8 | 43.7 | 43.0 | 29.0 | 33.1 |
| | SliceGPT | 65.7 | 0.0 | 54.2 | 40.0 | 3.7 | 44.8 | 33.0 | 21.2 | 25.7 |
| | ShortGPT | 59.4 | 5.6 | **75.5** | 46.8 | 11.9 | 43.1 | 38.8 | 24.0 | 29.5 |
| | Cus-Prun | **70.8** | **55.5** | 67.6 | **64.6** | **43.1** | **57.7** | **59.8** | **34.3** | **48.7** |

## 4.1 EXPERIMENT SETUP

**Benchmarks** Although `Cus-Prun` focuses on expert LLMs, which are evaluated on the specifically chosen dataset, we still assess its general capabilities to ensure minimal loss of overall performance. Specifically, we employ ARC-Challenge (Clark et al., 2018) (5-shots), GSM8K (Cobbe et al., 2021) (5-shots with CoT prompting (Wei et al., 2022)), and MMLU (Hendrycks et al., 2021) (5-shots) to represent models general capability. It's important to note that we utilize a generation task and implement CoT prompting method, approaches that has not been previously evaluated by existing pruning techniques (Song et al., 2024; Sharma et al., 2024; Yang et al., 2024b; Zhang et al., 2024).

**Baselines** We employ several pruning methods as the baseline that do not require post-training after pruning the model. (i) Dense represents the original model without pruning; (ii) LLM-Pruner (Ma et al., 2023) adopts structural pruning that selectively removes non-critical coupled structures based on gradient information;[2] (iii) SliceGPT (Ashkboos et al., 2024) replaces each weight matrix with a smaller dense matrix, reducing the embedding dimension of the network; (iv) ShortGPT (Men et al., 2024) directly deletes the redundant layers in LLMs based on their BI scores. Note that the pruning ratio is set to 25% for all methods and all models.

**Backbone Models** We choose 4 models that cover models from different series and different sizes, including Llama3-8B-Base (Dubey et al., 2024), Mistral-Nemo-Base-2407[3] (short as Mistral-12B), Llama2-13B-Base (Touvron et al., 2023), Llama3-70B-Base (Dubey et al., 2024).

## 4.2 MULTILINGUAL SETTING

**Benchmarks** We employ several conventional multilingual datasets for multilingual setting, which covers reasoning (MGSM (Shi et al., 2023), 5-shots), knowledge extraction (M3Exam (Zhang et al., 2023), 3-shots), understanding (XQuAD (Artetxe et al., 2020), 5-shots), and generation

---

[2]To ensure a fair comparison, we evaluate its performance before post-training, following Men et al. (2024).
[3]https://huggingface.co/mistralai/Mistral-Nemo-Base-2407

Table 2: Main Results of `Cus-Prun` on multidomain setting with a pruning ratio of 25%, where "general capability" is tested in English and averaged across several expert models. Results are expressed in accuracy (%) for all datasets. All models are base models.

| Model | Method | General Capability | | | | Expert Capability | | | | |
|---|---|---|---|---|---|---|---|---|---|---|
| | | ARC-c | GSM8K | MMLU | Avg. | MedMCQ | FinTQA | TSA | AMSA | Avg. |
| Llama3-8B | Dense | 70.7 | 58.3 | 63.1 | 64.1 | 51.8 | 23.9 | 67.1 | 95.9 | 59.8 |
| | LLMPrun. | 26.3 | 2.5 | 24.2 | 17.7 | 0.0 | 0.0 | **61.8** | 76.0 | 34.5 |
| | SliceGPT | 41.5 | 0.0 | 24.2 | 21.9 | 22.6 | 0.0 | 41.2 | 53.7 | 29.4 |
| | ShortGPT | 38.3 | 0.0 | 28.6 | 22.3 | 3.2 | 0.0 | 38.6 | 35.7 | 19.4 |
| | `Cus-Prun` | **63.7** | **39.1** | **57.8** | **53.5** | **42.9** | **20.6** | **61.8** | 87.6 | **53.2** |
| Mistral-12B | Dense | 82.6 | 68.5 | 50.4 | 67.2 | 54.6 | 26.6 | 69.4 | 92.4 | 60.8 |
| | LLMPrun. | 22.5 | 2.7 | 30.7 | 18.6 | 0.0 | 0.0 | 51.0 | 20.9 | 18.0 |
| | SliceGPT | 49.4 | 1.9 | 32.1 | 27.8 | 24.9 | 9.2 | 34.2 | 54.3 | 30.7 |
| | ShortGPT | 37.8 | 0.0 | 33.9 | 23.9 | 31.4 | 7.2 | 39.2 | 52.5 | 32.6 |
| | `Cus-Prun` | **67.3** | **47.8** | **45.7** | **53.6** | **47.9** | **25.1** | **67.3** | 83.7 | **56.0** |
| Llama2-13B | Dense | 50.3 | 31.4 | 53.4 | 45.1 | 25.2 | 0.0 | 42.7 | 84.1 | 38.0 |
| | LLMPrun. | 22.4 | 2.1 | 23.6 | 16.0 | 0.0 | 0.0 | 9.7 | 0.0 | 2.4 |
| | SliceGPT | 45.9 | 2.4 | 48.7 | 32.3 | 18.7 | 0.0 | 28.4 | 67.3 | 28.6 |
| | ShortGPT | 39.5 | 3.8 | 37.2 | 26.8 | 16.9 | 0.0 | 34.6 | **69.8** | 30.3 |
| | `Cus-Prun` | **48.6** | **21.2** | **50.5** | **40.1** | **25.6** | 0.0 | **38.5** | 68.3 | **33.1** |
| Llama3-70B | Dense | 84.1 | 82.7 | 78.8 | 81.9 | 72.1 | 55.3 | 83.6 | 96.2 | 76.8 |
| | LLMPrun. | **69.1** | 26.0 | 53.2 | 49.4 | 27.3 | 1.0 | 51.0 | 50.3 | 32.4 |
| | SliceGPT | 65.7 | 0.0 | 54.2 | 40.0 | 57.6 | 27.6 | 68.1 | 59.4 | 53.2 |
| | ShortGPT | 59.4 | 5.6 | **75.5** | 46.8 | 58.4 | 32.2 | 67.5 | 64.9 | 55.8 |
| | `Cus-Prun` | 68.0 | **51.2** | 66.4 | **61.9** | **68.2** | **43.9** | **81.4** | **87.8** | **70.3** |

(XLSum (Hasan et al., 2021), zero-shots). Furthermore, we cover three languages spanning a range from high-resource to low-resource including German (De), Chinese (Zh) and Thai (Th).

**Experiment Details** For multilingual setting, we can obtain two corpora: $C_{L_{\text{Exp}}} = \{(L_{\text{Exp}}, D, T) | D \in \mathbb{D}, T \in \mathbb{T}\}$ and $C_{(D,T)_{\text{Exp}}} = \{(L, (D, T)_{\text{Exp}}) | L \in \mathbb{L}\}$. The first corpus contains samples in a specific language across various domains and tasks, while the second corpus contains samples from a specific domain-task combination in other languages, i.e., the target dataset in other languages. Specifically, for $C_{L_{\text{Exp}}}$ we employ Wikipedia[4] to construct language-specific corpus covering various domains and tasks. For $C_{(D,T)_{\text{Exp}}}$, we employ the corresponding datasets in English, including GSM8K (Cobbe et al., 2021) for MGSM, the English split of M3Exam[5] for M3Exam, SQuAD (Rajpurkar, 2016) for XQuAD, and XSum (Narayan et al., 2018) for XLSum. More detailed experiment settings are explained in Appendix A.1.1.

**Main Results** Table 1 shows the performance of `Cus-Prun` on multilingual datasets, which is the average performance across languages and detailed results in each language is shown in Table 5, Table 6 and Table 7 in Appendix A.2. We find that `Cus-Prun` consistently outperforms other pruning methods in obtaining expert models for multilingual settings while maintaining its general capability. Specifically, for expert capabilities, `Cus-Prun` achieves a score of 38.9 on Llama3-8B, while other pruning methods achieve at most 15.5. The scores are 34.3 for Mistral-12B, 24.3 for Llama2-13B, and 48.7 for Llama3-70B, all significantly higher than those of other pruning methods, which achieve at most 18.0, 15.5 and 29.5 for three models respectively. For general capabilities, `Cus-Prun` also performs better than other baselines. The cases are the same for other models.

Moreover, the performance improvement of `Cus-Prun` is more pronounced in tasks requiring generation rather than direct classification. Specifically, `Cus-Prun` achieves a score of 30.1 on MGSM for Llama3-8B, with scores of 39.2, 26.2, and 57.7 for Mistral-12B, Llama2-13B, and Llama3-70B, respectively. In contrast, other pruning methods almost entirely lose the ability to generate reasoning thoughts, achieving accuracy close to 0 for models other than Llama3-70B.

---

[4] `https://huggingface.co/datasets/wikimedia/wikipedia`
[5] M3Exam is language-specific and does not utilize a translated parallel corpus.

Table 3: Main Results of `Cus-Prun` on multitask setting with a pruning ratio of 25%, where "general capability" is tested in English and averaged across expert models. Results are expressed in Rouge-L for MedSum and AMSum and in accuracy (%) for others. All models are base models.

| Model | Method | General Capability | | | | Expert Capability | | | |
|-------|--------|------|------|------|------|--------|-------|----------|------|
| | | ARC-c | GSM8K | MMLU | Avg. | MedSum | AMSum | AMContFact | Avg. |
| Llama3-8B | Dense | 70.7 | 58.3 | 63.1 | 64.1 | 76.6 | 16.2 | 78.2 | 57.0 |
| | LLMPrun. | 26.3 | 2.5 | 24.2 | 17.7 | 62.2 | **21.8** | **80.0** | **54.7** |
| | SliceGPT | 41.5 | 0.0 | 24.2 | 21.9 | 7.3 | 2.9 | 51.3 | 20.5 |
| | ShortGPT | 38.3 | 0.0 | 28.6 | 22.3 | 4.1 | 4.8 | 43.8 | 17.6 |
| | Cus-Prun | **63.2** | **40.1** | **54.3** | **52.5** | 68.4 | 12.8 | 75.5 | 52.2 |
| Mistral-12B | Dense | 82.6 | 68.5 | 50.4 | 67.2 | 88.7 | 3.0 | 78.6 | 56.4 |
| | LLMPrun. | 22.5 | 2.7 | 30.7 | 18.6 | 59.3 | 0.5 | 2.8 | 20.9 |
| | SliceGPT | 49.4 | 1.9 | 32.1 | 27.8 | 27.4 | 1.3 | 36.3 | 21.7 |
| | ShortGPT | 37.8 | 0.0 | 33.9 | 23.9 | 26.2 | 0.2 | 42.7 | 23.0 |
| | Cus-Prun | **68.1** | **42.7** | **42.2** | **51.0** | **83.5** | **3.4** | 72.8 | 50.9 |
| Llama2-13B | Dense | 50.3 | 31.4 | 53.4 | 45.1 | 70.0 | 7.4 | 44.3 | 40.6 |
| | LLMPrun. | 22.4 | 2.1 | 23.6 | 16.0 | 21.6 | 4.8 | 0.0 | 8.8 |
| | SliceGPT | 45.9 | 2.4 | **48.7** | 32.3 | 24.5 | 4.9 | 32.9 | 20.8 |
| | ShortGPT | 39.5 | 3.8 | 37.2 | 26.8 | 23.8 | 5.2 | 39.1 | 22.7 |
| | Cus-Prun | **48.2** | **20.6** | **48.7** | **39.2** | **64.5** | **6.7** | **42.9** | **38.0** |
| Llama3-70B | Dense | 84.1 | 82.7 | 78.8 | 81.9 | 84.2 | 17.3 | 81.8 | 61.1 |
| | LLMPrun. | **69.1** | 26.0 | 53.2 | 49.4 | 10.2 | 13.7 | 20.6 | 14.8 |
| | SliceGPT | 65.7 | 0.0 | 54.2 | 40.0 | 58.0 | 14.2 | 68.3 | 46.8 |
| | ShortGPT | 59.4 | 5.6 | **75.5** | 46.8 | 59.6 | 13.9 | 65.8 | 46.4 |
| | Cus-Prun | 66.3 | **52.9** | 65.9 | **61.7** | **80.4** | **15.7** | **77.5** | **57.9** |

## 4.3 MULTIDOMAIN SETTING

**Benchmarks**   For the multidomain setting, we employ several domain-specific datasets, including medical domain multiply choices questions (MedMCQ (Pal et al., 2022), 3-shots), finance domain table question-answering (FinTQA (Chen et al., 2021), 8-shots), social media domain sentiment analysis (TSA (Kharde & Sonawane, 2016), 3-shots), and e-commerce domain sentiment analysis (AMSA (Zhang et al., 2015), 3-shots). Moreover, in multidomain setting, our focus is exclusively on the English language. Detailed experiment settings are explained in A.1.2.

**Main Results**   Table 2 shows the performance of `Cus-Prun` on multidomain setting. We find that `Cus-Prun` consistently outperforms other pruning methods in both expert and general capabilities. For expert capabilities, `Cus-Prun` achieves a score of 53.2 on Llama3-8B, while other pruning methods achieve at most 34.5. The scores are 56.0 for Mistral-12B, 33.1 for Llama2-13B, and 70.3 for Llama3-70B, all significantly higher than those of other pruning methods, which achieve at most 32.6, 30.3 and 55.8 for three models respectively.

## 4.4 MULTITASK SETTING

**Benchmarks**   For the multitask setting, we employ several task-specific datasets, including the medical summarization task (MeQSum (Abacha & Demner-Fushman, 2019), 3-shots), summarization task in e-commerce (Amazon Summary (Wang et al., 2022; Brüel-Gabrielsson et al., 2024), 3-shots), counterfactual task in e-commerce (Amazon Counterfactual (O'Neill et al., 2021), 3-shots). Similarly, in multitask setting scenarios, our focus is exclusively on the English language. Detailed experiment settings are explained in A.1.3.

**Main Results**   Table 3 shows the performance of `Cus-Prun` on multitask setting. We find that except for LLM-Pruner under Llama3-8B, `Cus-Prun` outperforms other pruning methods in both expert and general capabilities. For expert capabilities, `Cus-Prun` achieves a score of 50.9 on Mistral-12B, while other pruning methods achieve at most 23.0. The scores are 38.0 for Llama2-13B, and 57.9 for Llama3-70B, all significantly higher than those of other pruning methods, which achieve at most 22.7 and 46.8 for the two models respectively.

Table 4: Performance of Chinese-Medical expert model on MCQ task.

| Method | General | CMExam |
|--------|---------|--------|
| Dense | 59.3 | 50.6 |
| LLM-Pruner | 18.6 | 25.0 |
| SliceGPT | 27.8 | 26.9 |
| ShortGPT | 23.9 | 23.7 |
| Cus-Prun | **52.4** | **48.7** |

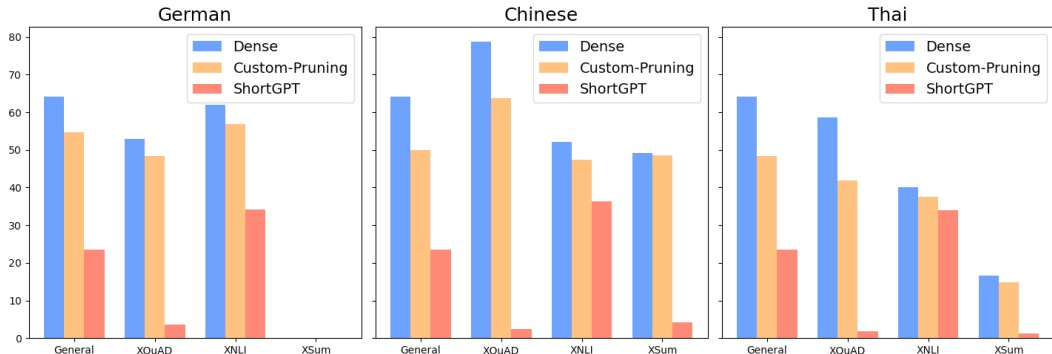

Figure 3: Chinese Medical LLM QA performance. Numbers are quality on the **whole** testset evaluated by GPT4.

Figure 4: Performance of Cus-Prun in obtaining language-specific models.

## 5 ADAPTIVE CUSTOM PRUNING ASSESSMENT

In this section, we evaluate the generality of Cus-Prun in dynamic scenarios, including specific expert models in two and one dimensions, as described in Section 2.2.

### 5.1 TWO DIMENSIONS SPECIFIC EXPERT MODEL

**Experiment Settings**  We use a Chinese-Medical LLM as a concrete example of a two-dimensional expert model, capable of performing various medical tasks in Chinese. We adopt Mistral-12b as the backbone model and utilize corpus from Wikipedia for Chinese content and general medical corpus for medical knowledge. The performance of the Chinese-Medical expert model is primarily evaluated on two datasets: CMExam (Liu et al., 2023) (5-shots), a Chinese medical multiple-choice question dataset, and HuatuoQA (Li et al., 2023a), a Chinese medical question-answering dataset. We assess the performance on CMExam using accuracy metrics, while the performance on HuatuoQA is more challenging to evaluate quantitatively. For the latter, we sample a sub-testset of size 100 and use GPT-4 as the evaluator, which assigns a score from 0 to 5, representing its quality from low to high.

**Main Results**  Table 4 presents the performance of the Chinese-Medical LLM on CMExam and its general capabilities. Our results indicate that the expert model pruned using Cus-Prun outperforms models obtained through other pruning methods. Specifically, Cus-Prun achieves a score of 48.7 on CMExam, while its general capability score is 52.4. These results compare favorably to the dense model, which scores 50.6 on CMExam and 59.3 on general capabilities. On the contrary, other pruning methods nearly lose the general and specific capabilities. Furthermore, Figure 3 shows a concrete example of Chinese-Medical LLM performance on medical question-answering. We find that Cus-Prun can produce smaller expert models that maintain their expert capabilities, as demonstrated by its performance score of 2.9 compared to 3.2 for the dense model.

### 5.2 ONE DIMENSION SPECIFIC EXPERT MODEL

**Experiment Settings**  For evaluating the pruning method under a one-dimensional expert model setting, we focus on language-specific pruning, showing how to transform a dense model into

language-specific variants. We consider three linguistically diverse languages: German, Chinese, and Thai, and conduct experiments based on the Llama3-8b model. To identify language-specific (while domain- and task-agnostic) neurons, we employ a diverse range of corpora, including Wikipedia, MGSM, and M3Exam, ensuring coverage of various domains and tasks. The effectiveness of our pruning technique is then evaluated using three held-out multilingual datasets including XQuAD (Artetxe et al., 2020), XNLI (Conneau et al., 2018), and XSum (Narayan et al., 2018).

**Main Results** Figure 4 illustrates the performance of language-specific models using `Cus-Prun`. By pruning $25\%$ of the neurons from the original model, `Cus-Prun` not only retains general performance but also preserves language-specific capabilities. For instance, the German-specific model scores $54.7$ in general capabilities, $48.3$ on XQuAD, and $56.8$ on XNLI, compared to the dense model's scores of $64.1$, $52.9$, and $62.0$, respectively. This trend is consistent for Chinese and Thai models as well. In contrast, ShortGPT struggles to maintain the model's capabilities, particularly in XQuAD and XSUm, which require generative abilities rather than simple classification.

## 6 RELATED WORK

**LLM Compression** Given the high costs associated with training, inferencing, and tuning LLMs, many studies explore methods to compress the model to conserve computing resources, including model compression (Zhu et al., 2023), quantization (Xu et al., 2023; Dettmers et al., 2024; Lin et al., 2024; Li et al., 2024), and pruning (Wang et al., 2019). In the context of pruning, sparsity serves as a structural pruning (Li et al., 2022; 2023c; Kurz et al., 2024; Zhao et al., 2024a; Huang et al., 2024), which doesn't save computing resources but leverages GPU calculation properties for acceleration. In addition, some works develop unstructuraled pruning methods aimed at reducing model parameters while maintaining general performance. They either employ extensive post-training (Ma et al., 2023; Xia et al., 2024; Muralidharan et al., 2024), nor adopt coarse-grained pruning method at structure such as approximating all parameters (Zhao et al., 2024a), removing entire layers (Men et al., 2024), or eliminating network structures (Zhang et al., 2024). However, they fail to capture the model's expert capability thus fail to be applied to more specific downstream scenarios.

**Customizing Model** The rapid evolution of LLMs has led to a growing need for customization to meet specific requirements across various fields. Language-specific models are being developed to address unique linguistic needs (Cui et al., 2023; Yang et al., 2024b), while domain-specific models cater to specialized areas like healthcare and software development (Li et al., 2023a; Roziere et al., 2023; Li et al., 2023b). Task-specific models further enhance performance for particular applications (Azerbayev et al., 2024; Alves et al., 2024). However, correctly customizing these models requires extensive fine-tuning with a tailored training corpus. This challenge highlights the need for efficient methods to acquire and refine expert models, ensuring LLMs can be adapted effectively to meet diverse industry demands.

## 7 CONCLUSION

LLMs offer impressive capabilities but come with substantial computational costs. Efficient pruning of redundant parameters is crucial for conserving resources and improving speed, especially for users requiring specialized models for specific tasks. While current pruning methods often demand extensive post-training or lack precision, our proposed method, `Cus-Prun`, creates smaller expert models without post-training. By mapping models along "language," "domain," and "task" dimensions and pruning irrelevant neurons, `Cus-Prun` achieves efficient expert model creation in a finer-grained manner. Experimental results demonstrate that `Cus-Prun` consistently outperforms existing techniques on three-dimensional specific models. Furthermore, `Cus-Prun` can be tailored to more realistic scenarios by targeting just one or two dimensions, such as language-domain or language-specific models, experimentally outperforming other pruning methods in these contexts as well.

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

## A    APPENDIX

### A.1    EXPERIMENTS DETAILED SETTINGS

#### A.1.1    MULTILINGUAL SETTINGS

**Experiment Details**    Hyperparameters, including the sizes of $C_{L_{\text{Exp}}}$ and $C_{(D,T)_{\text{Exp}}}$, are determined using the validation set of the XLSum dataset and then applied to testsets in other multilingual datasets. Furthermore, accuracy is the metric used for ARC-c, GSM8K, MMLU, MGSM, M3Exam, and XQuAD, while Rouge-L (Lin, 2004) is used for XLSum.

#### A.1.2    MULTIDOMAIN SETTINGS

**Settings**    For multidomain setting, we can obtain two corpora: $C_{D_{\text{Exp}}} = \{(L, D_{\text{Exp}}, T)|L \in \mathbb{L}, T \in \mathbb{T}\}$ and $C_{(L,T)_{\text{Exp}}} = \{(D, (L, T)_{\text{Exp}})|D \in \mathbb{D}\}$. The first corpus contains samples in a specific domain across various languages and tasks, while the second corpus contains samples from a specific language-task combination across different domains, i.e., the target dataset in other domains. Specifically, for $C_{D_{\text{Exp}}}$ we employ specific domain corpus, including English split of medical corpus (García-Ferrero et al., 2024) for medical domain, general finance corpus for finance domain[6], general Twitter corpus (Kharde & Sonawane, 2016), and English split of Amazon corpus (Keung et al., 2020). For $C_{(L,T)_{\text{Exp}}}$, we employ the corresponding datasets in general domains, including CommonsenseQA (Talmor et al., 2019) for MedMCQ, open table question-answering OTT-QA (Chen et al., 2020) for FinTQA, general sentiment analysis (Attia et al., 2018) for TSA and AMSA.

**Experiment Details**    Hyperparameters, including the sizes of $C_{D_{\text{Exp}}}$ and $C_{(L,T)_{\text{Exp}}}$, are determined using the validation set of the Amazon sentiment analysis dataset and then applied to testsets in other multidomain datasets. Furthermore, accuracy is the metric used for all datasets.

#### A.1.3    MULTITASK SETTINGS

**Settings**    For multitask setting, we can obtain two corpora: $C_{T_{\text{Exp}}} = \{(L, D, T_{\text{Exp}})|L \in \mathbb{L}, D \in \mathbb{D}\}$ and $C_{(L,D)_{\text{Exp}}} = \{(T, (L, S)_{\text{Exp}})|T \in \mathbb{T}\}$. The first corpus contains samples in a specific task across various languages and domains, while the second corpus contains samples from a specific language-domain combination across different tasks, i.e., the target dataset in other tasks. Specifically, for $C_{T_{\text{Exp}}}$ we employ specific task corpus, including XSum corpus (Abacha & Demner-Fushman, 2019) for summarization task, general conterfact corpus[7] for counterfactual task. For $C_{(L,D)_{\text{Exp}}}$, we employ the corresponding datasets in other tasks, including MedQCQ (Pal et al., 2022) for MedSum, AMSA (Zhang et al., 2015) for AMSum and AMContFact.

**Experiment Details**    Hyperparameters, including the sizes of $C_{T_{\text{Exp}}}$ and $C_{(L,D)_{\text{Exp}}}$, are determined using the validation set of the Amazon counterfactual dataset and then applied to testsets in other multitask setting datasets. Furthermore, accuracy is the metric used for ARC-c, GSM8K, MMLU, and AMContFact, while Rouge-L (Lin, 2004) is used for MedSum and AMSum.

### A.2    DETAILED RESULTS FOR MULTILINGUAL

---

[6]https://huggingface.co/datasets/gbharti/finance-alpaca
[7]https://huggingface.co/datasets/azhx/counterfact-easy

| Model | Method | General Capability | | | | Expert Capability | | | | |
|-------|--------|------|------|------|------|------|--------|-------|-------|------|
| | | ARC-c | GSM8K | MMLU | Avg. | MGSM | M3Exam | XQuAD | XLSum | Avg. |
| Llama3-8B | Dense | 70.7 | 58.3 | 63.1 | 64.1 | 44.8 | - | 52.9 | - | 48.8 |
| | LLMPrun. | 26.3 | 2.5 | 24.2 | 17.7 | 0.0 | - | 11.0 | - | 5.5 |
| | SliceGPT | 41.5 | 0.0 | 24.2 | 21.9 | 0.0 | - | 9.8 | - | 4.9 |
| | ShortGPT | 38.3 | 0.0 | 28.6 | 22.3 | 0.0 | - | 0.0 | - | 0.0 |
| | Cus-Prun | 61.4 | 38.9 | 54.5 | **51.6** | 32.8 | - | 49.6 | - | **41.2** |
| Mistral-12B | Dense | 82.6 | 68.5 | 50.4 | 59.3 | 56.8 | - | 41.2 | - | 49.0 |
| | LLMPrun. | 22.5 | 2.7 | 30.7 | 18.6 | 2.4 | - | 13.4 | - | 7.9 |
| | SliceGPT | 49.4 | 1.9 | 32.1 | 27.8 | 0.8 | - | 15.5 | - | 8.2 |
| | ShortGPT | 37.8 | 0.0 | 33.9 | 23.9 | 3.6 | - | 20.3 | - | 12.0 |
| | Cus-Prun | 64.6 | 39.7 | 43.2 | **49.2** | 31.6 | - | 35.9 | - | **33.8** |
| Llama2-13B | Dense | 50.3 | 31.4 | 53.4 | 45.1 | 24.4 | - | 40.3 | - | 32.3 |
| | LLMPrun. | 22.4 | 2.1 | 23.6 | 16.0 | 2.0 | - | 5.7 | - | 3.9 |
| | SliceGPT | 45.9 | 2.4 | 48.7 | 32.3 | 3.6 | - | 18.1 | - | 10.9 |
| | ShortGPT | 39.5 | 3.8 | 37.2 | 26.8 | 2.8 | - | 27.2 | - | 15.0 |
| | Cus-Prun | 47.6 | 19.8 | 49.9 | **39.1** | 18.4 | - | 31.7 | - | **25.0** |
| Llama3-70B | Dense | 84.1 | 82.7 | 78.8 | 81.9 | 74.8 | - | 58.2 | - | 66.5 |
| | LLMPrun. | 69.1 | 26.0 | 53.2 | 49.4 | 18.0 | - | 27.3 | - | 22.7 |
| | SliceGPT | 65.7 | 0.0 | 54.2 | 40.0 | 0.0 | - | 17.3 | - | 8.7 |
| | ShortGPT | 59.4 | 5.6 | 75.5 | 46.8 | 9.6 | - | 31.5 | - | 20.6 |
| | Cus-Prun | 66.8 | 59.3 | 69.1 | **65.1** | 48.2 | - | 53.9 | - | **51.1** |

Table 5: Germany.

| Model | Method | General Capability | | | | Specific Capability | | | | |
|-------|--------|------|------|------|------|------|--------|-------|-------|------|
| | | ARC-c | GSM8K | MMLU | Avg. | MGSM | M3Exam | XQuAD | XLSum | Avg. |
| Llama3-8B | Dense | 70.7 | 58.3 | 63.1 | 64.1 | 43.6 | 55.1 | 78.7 | 49.1 | 56.6 |
| | LLMPrun. | 26.3 | 2.5 | 24.2 | 17.7 | 2.4 | 23.6 | 21.3 | 32.8 | 20.0 |
| | SliceGPT | 41.5 | 0.0 | 24.2 | 21.9 | 0.0 | 17.4 | 23.5 | 8.3 | 12.3 |
| | ShortGPT | 38.3 | 0.0 | 28.6 | 22.3 | 0.0 | 28.3 | 0.0 | 3.1 | 7.9 |
| | Cus-Prun | 60.5 | 25.7 | 49.4 | **45.2** | 36.0 | 44.7 | 65.6 | 46.3 | **48.2** |
| Mistral-12B | Dense | 82.6 | 68.5 | 50.4 | 59.3 | 53.2 | 47.8 | 62.2 | 33.0 | 49.1 |
| | LLMPrun. | 22.5 | 2.7 | 30.7 | 18.6 | 2.8 | 30.7 | 31.8 | 32.6 | 24.5 |
| | SliceGPT | 49.4 | 1.9 | 32.1 | 27.8 | 1.6 | 26.4 | 28.3 | 10.8 | 16.8 |
| | ShortGPT | 37.8 | 0.0 | 33.9 | 23.9 | 4.4 | 28.2 | 29.1 | 7.2 | 17.2 |
| | Cus-Prun | 68.3 | 43.2 | 39.5 | **50.3** | 38.4 | 40.7 | 50.6 | 30.3 | **40.0** |
| Llama2-13B | Dense | 50.3 | 31.4 | 53.4 | 45.1 | 21.6 | 36.5 | 59.8 | 35.3 | 38.3 |
| | LLMPrun. | 22.4 | 2.1 | 23.6 | 16.0 | 1.2 | 23.3 | 3.8 | 25.1 | 13.4 |
| | SliceGPT | 45.9 | 2.4 | 48.7 | 32.3 | 4.8 | 24.5 | 28.4 | 11.2 | 17.2 |
| | ShortGPT | 39.5 | 3.8 | 37.2 | 26.8 | 4.4 | 22.9 | 24.6 | 13.7 | 16.4 |
| | Cus-Prun | 48.6 | 20.7 | 51.9 | **40.4** | 14.8 | 28.2 | 47.3 | 34.4 | **31.2** |
| Llama3-70B | Dense | 84.1 | 82.7 | 78.8 | 81.9 | 68.4 | 76.1 | 81.3 | 55.3 | 70.3 |
| | LLMPrun. | 69.1 | 26.0 | 53.2 | 49.4 | 16.8 | 47.5 | 56.1 | 41.3 | 40.4 |
| | SliceGPT | 65.7 | 0.0 | 54.2 | 40.0 | 6.4 | 48.3 | 42.2 | 29.3 | 31.6 |
| | ShortGPT | 59.4 | 5.6 | 75.5 | 46.8 | 12.4 | 45.5 | 44.6 | 36.1 | 34.7 |
| | Cus-Prun | 72.3 | 48.5 | 65.2 | **62.0** | 40.8 | 61.7 | 66.9 | 51.6 | **55.3** |

Table 6: Chinese.

| Model | Method | General Capability | | | | Specific Capability | | | | |
|---|---|---|---|---|---|---|---|---|---|---|
| | | ARC-c | GSM8K | MMLU | Avg. | MGSM | M3Exam | XQuAD | XLSum | Avg. |
| Llama3-8B | Dense | 70.7 | 58.3 | 63.1 | 64.1 | 35.2 | 43.0 | 58.7 | 16.7 | 38.4 |
| | LLMPrun. | 26.3 | 2.5 | 24.2 | 17.7 | 0.8 | 24.4 | 8.4 | 13.5 | 11.8 |
| | SliceGPT | 41.5 | 0.0 | 24.2 | 21.9 | 0.0 | 12.3 | 16.6 | 8.7 | 9.4 |
| | ShortGPT | 38.3 | 0.0 | 28.6 | 22.3 | 0.0 | 25.4 | 0.0 | 2.3 | 6.9 |
| | Cus-Prun | 58.9 | 31.2 | 52.4 | **47.5** | 21.6 | 38.3 | 42.6 | 16.8 | **29.8** |
| Mistral-12B | Dense | 82.6 | 68.5 | 50.4 | 59.3 | 45.2 | 39.9 | 44.1 | 17.8 | 36.8 |
| | LLMPrun. | 22.5 | 2.7 | 30.7 | 18.6 | 1.2 | 24.8 | 11.9 | 13.7 | 12.9 |
| | SliceGPT | 49.4 | 1.9 | 32.1 | 27.8 | 0.0 | 23.8 | 8.4 | 4.7 | 12.3 |
| | ShortGPT | 39.5 | 3.8 | 37.2 | 26.8 | 0.8 | 25.7 | 4.7 | 2.8 | 8.5 |
| | Cus-Prun | 68.2 | 35.8 | 47.6 | **50.5** | 32.8 | 37.7 | 35.6 | 15.9 | **30.5** |
| Llama2-13B | Dense | 50.3 | 31.4 | 53.4 | 45.1 | 6.4 | 24.3 | 28.3 | 14.5 | 18.4 |
| | LLMPrun. | 22.4 | 2.1 | 23.6 | 16.0 | 0.0 | 22.3 | 1.8 | 10.2 | 8.6 |
| | SliceGPT | 45.9 | 2.4 | 48.7 | 32.3 | 0.0 | 26.2 | 23.7 | 8.6 | 14.6 |
| | ShortGPT | 39.5 | 3.8 | 37.2 | 26.8 | 0.0 | 23.1 | 22.3 | 8.9 | 13.6 |
| | Cus-Prun | 47.8 | 20.9 | 50.7 | **39.8** | 4.8 | 24.2 | 23.6 | 13.8 | **16.6** |
| Llama3-70B | Dense | 84.1 | 82.7 | 78.8 | 81.9 | 65.2 | 66.1 | 67.8 | 17.8 | 54.2 |
| | LLMPrun. | 69.1 | 26.0 | 53.2 | 49.4 | 15.6 | 39.9 | 29.8 | 16.6 | 25.5 |
| | SliceGPT | 65.7 | 0.0 | 54.2 | 40.0 | 4.8 | 41.3 | 39.6 | 13.2 | 24.7 |
| | ShortGPT | 59.4 | 5.6 | 75.5 | 46.8 | 13.7 | 40.7 | 40.4 | 11.9 | 26.7 |
| | Cus-Prun | 73.3 | 58.7 | 68.4 | **66.8** | 40.4 | 53.6 | 58.5 | 16.9 | **42.4** |

Table 7: Thai.