# OpenReview forum: "From General to Expert: Custom Pruning LLMs Across Language, Domain, and Task"
_ICLR.cc/2025/Conference — ICLR 2025 Conference Withdrawn Submission_

### Official Review · Reviewer_FwM7 · 2024-11-01

**Soundness:** 2
**Presentation:** 2
**Contribution:** 1
**Rating:** 3
**Confidence:** 5

**Summary:**

The paper presents a new LLM pruning approach (Cus-Prun) that prunes redundant parameters to improve memory efficiency. In particular, the authors take "language", "domain" and "task" as 3 dimensions into account and collect corresponding corpus to identify irrelevant parameters and zero out them. Experimental results on several benchmarks demonstrate the effectiveness of the approach.

**Strengths:**

The paper is easy to understand, and the idea is quite straightforward and clear.

**Weaknesses:**

The novelty of the paper is quite limited, and the result of the paper is questionable. In particular:
- There is nothing new on how to identify the relevance of examples, so I regard the main novelty as the split of language, domain and task corpus. Such split is very heuristic without clear motivation. What if we just let users provide a corpus mixture that they care about, and apply any pruning technique on such corpus. Based on the idea from the paper, such user-provided corpus should contain even more than 3 pre-defined dimensions, for example, 1) special instructions, 2) special formats, etc. Then why bother splitting them into 3 hard-coded dimensions and follow the framework?
- What is the benefit of pruning 25% parameters at the cost of 10+% drop in capacity? Without special software and hardware design, pruning 25% parameters does not directly translate to any memory saving or inference speed up.
- If I understand correctly, pruning 25% of parameters of a 70B model translates to a 50B+ parameter model. Can the authors compare the performance with even a <50B dense model?

**Questions:**

Please see the weaknesses part.

---

### Official Review · Reviewer_jZX8 · 2024-11-02

**Soundness:** 2
**Presentation:** 2
**Contribution:** 2
**Rating:** 3
**Confidence:** 5

**Summary:**

This work proposes Cus-Prun, which creates smaller expert models from pre-trained LLMs without post-training. Experimental results demonstrate that Cus-Prun outperforms existing techniques on three-dimensional ("language," "domain," and "task”) specific models.

**Strengths:**

- The proposed task-specific pruning without post-training setting is novel.
- Extensive experiments across several tasks and models are included, validating the effectiveness of the proposed method.

**Weaknesses:**

- Algorithm 1 is unclear. The authors should point out how the “irrelevant neurons” are identified.
- This work aims “to optimize computing resources and increase inference speed.” However, the experimental evaluation does not contain any results about efficiency (e.g., inference speedup).
- The ablation study in this work is limited. For example, the impact of the pre-defined threshold ϵ should be further studies.

**Questions:**

- Figure 1 looks confusing to me. For example, what do the black and gray cross marks mean?

---

### Official Review · Reviewer_SyzM · 2024-11-03

**Soundness:** 3
**Presentation:** 3
**Contribution:** 3
**Rating:** 6
**Confidence:** 3

**Summary:**

This paper introduces a novel framework for structured pruning named Cus-Prun, which creates "expert models", or pruned networks that are tailored for a combination of specific languages (L), tasks (T) and/or domains (D). The framework is based on the assumption that targeting specific languages, tasks and domains creates more opportunities for pruning, and can yield better accuracy at the same pruning ratio compared to more general frameworks such as ShortGPT or SliceGPT. The pruning itself is performed by identifying redundant or irrelevant neurons for different (L, T and D) combinations - specifically, the paper adopts an Oracle-based approach, where importance of a neuron is determined by the change in model accuracy when it is driven to zero (compared to original dense network). Cus-Prun is evaluated across a range of different (L, T, D) combinations and compared with both recent state-of-the-art LLMs and general pruning approaches.

**Strengths:**

* The paper is reasonably well-written, with core ideas and concepts clearly explained.
* Creating strong networks/sub-networks for specific domains, tasks and languages is an important problem.

**Weaknesses:**

* The evaluation includes comparisons to other LLMs and pruning methods, but not to networks tailored for specific languages, domains, and tasks. While I understand that it may be hard to obtain open LLMs for each combination, without this comparison, it’s hard to understand if a much smaller customized LLM potentially outperforms the pruned network(s) obtained using Cus-Prun. A simple way to do this comparison might be to take either (1) the original baseline LLM, or (2) a 25% smaller LLM (pruning ratio used in the paper) and then perform PEFT (using LoRA for example) on the corresponding dataset.
* Oracle-based pruning is not really novel.

**Questions:**

* What is the overhead of oracle-based importance estimation? This needs to be clearly specified. The number of neurons in modern LLMs is very large, and measuring change in loss/accuracy by holding out each one individually could be potentially very high.
* It’s not clear if the accuracy improvements in Tables 1-3 are due to the pruning method or the customization (especially considering the strong accuracy numbers shown for “general capability”). Would it be possible to apply the same method to obtain general pruned models? This would involve using oracle-based pruning on perhaps a randomized subset of data across languages, domains and tasks. Comparing the performance of such a model to the customized ones presented in the paper will clarify this aspect.
* How does Cus-Prun perform at higher pruning ratios? For example, 33% or 50%? Since the resulting models are domain/task-specific, it seems plausible that higher compression ratios would be achievable.
* How does instruct-tuning affect model performance compared to dense or other pruned models?

---

### Official Review · Reviewer_BCPY · 2024-11-03

**Soundness:** 3
**Presentation:** 3
**Contribution:** 2
**Rating:** 5
**Confidence:** 3

**Summary:**

This paper aims to address the limitations of existing pruning techniques that typically focus on general capabilities and often employ coarse-grained pruning approaches, and sometimes require extensive post-training after pruning. The authors propose a more targeted approach, aiming to develop expert models by selectively pruning neurons that do not contribute to the model's performance in specific areas. This method involves identifying irrelevant neurons for each dimension (language, domain, task) by evaluating the impact of their removal on the model’s output when processing the corresponding corpus. Experimental results demonstrate minimal loss in both expert and general capabilities across various models from different families and sizes, positioning expert models effectively along the “language”, “domain”, and “task” dimensions.

**Strengths:**

1. The idea of positioning an expert model along specific dimensions such as language, domain, and task is innovative and offers enhanced flexibility for model adaptation. This approach allows for more specialized and efficient models tailored to specific tasks or domains.
2. The experiments conducted are comprehensive, evaluating the performance of the proposed method across a variety of scenarios. Additionally, the inclusion of different model families and sizes in the experimental setup adds robustness to the findings, showcasing the method’s versatility.
3. The paper is well-written and easy to follow.

**Weaknesses:**

The paper would benefit from an analysis comparing the FLOPs before and after pruning. This would provide quantitative proof of the efficiency gains from the pruning process, substantiating the method’s effectiveness in reducing computational overhead while maintaining performance.

**Questions:**

In Chapter 2, the description of "the l-th neuron in layer i" raises questions about how a neuron is precisely defined within this context. Does this refer to all the parameters within a particular layer (e.g., FFN, MLP), or is it specific to certain parameters? Clarification on this would aid in understanding the granularity of the pruning process.

---

### Note · Authors · 2024-12-15

I have read and agree with the venue's withdrawal policy on behalf of myself and my co-authors.